# Gingival Crevicular Blood as a Potential Screening Tool: A Cross Sectional Comparative Study

**DOI:** 10.3390/ijerph17207356

**Published:** 2020-10-09

**Authors:** Biagio Rapone, Elisabetta Ferrara, Luigi Santacroce, Skender Topi, Ilaria Converti, Antonio Gnoni, Antonio Scarano, Salvatore Scacco

**Affiliations:** 1Department of Basic Medical Sciences, Neurosciences and Sense Organs, “Aldo Moro” University of Bari, 70124 Bari, Italy; gnoniantonio@gmail.com; 2Complex Operative Unit of Odontostomatology, Hospital S.S. Annunziata, 66100 Chieti, Italy; igieneeprevenzione@gmail.com; 3Ionian Department (DJSGEM), Microbiology and Virology Lab., “Aldo Moro” University of Bari, 70124 Bari, Italy; luigi.santacroce@uniba.it; 4Department of Clinical Disciplines, “A. Xhuvani” Elbasan University, 3001 Elbasan, Albania; skender.topi@uniel.edu.al; 5Department of Emergency and Organ Transplantation, Division of Plastic and Reconstructive Surgery, “Aldo Moro” University of Bari, 70124 Bari, Italy; ilaria.converti@gmail.com; 6Department of Oral Science, Nano and Biotechnology and CeSi-Met University of Chieti-Pescara, 66100 Chieti, Italy; ascarano@unich.it

**Keywords:** screening, diabetes, periodontal disease, periodontitis, glucose measurement

## Abstract

Background: Diabetes is known to be one of the major global epidemic diseases, significantly associated with mortality and morbidity worldwide, conferring a substantial burden to the health care system. The epidemiological transition of this chronic disease tends to worsen unless preventive health strategies are implemented. Appropriate screening devices and standardized methods are crucial to prevent this potentially inauspicious life condition. Currently, the glucometer is the conventional device employed for blood glucose level determination that outputs the blood glucose reading. Glucometer performed in the dental office may be an important device in screening diabetes, so it can be addressed during a periodontal examination. Because gingival blood is a useful source to detect the glucose level, the focus is placed on the opportunity that might provide valuable diagnostic information. This study aimed to compare gingival crevicular blood with finger-stick blood glucose measurements using a self-monitoring glucometer, to evaluate whether gingival crevicular blood could be an alternative to allow accurate chairside glucose testing. Methods: A cross-sectional comparative study was performed among a 31–67-year-old population. Seventy participants with diagnosed type 2 diabetes and seventy healthy subjects, all with positive bleeding on probing, were enrolled. The gingival crevicular blood was collected using a glucometer to estimate the blood glucose level and compared with finger-stick blood glucose level. Results: The mean capillary blood glucose and gingival crevicular blood levels from all samples were, respectively, 160.42 ± 31.31 mg/dL and 161.64 ± 31.56 mg/dL for diabetic participants and 93.51 ± 10.35 mg/dL and 94.47 ± 9.91 mg/dL for healthy patients. In both groups, the difference between gingival crevicular blood and capillary blood glucose levels was non-significant (*P* < 0.05). The highly significant correlation between capillary blood glucose and gingival crevicular blood (*r* = 0.9834 for diabetic patients and *r* = 0.8153 for healthy participants) in both the groups was found. Conclusions: Gingival crevicular blood test was demonstrated as a feasible and useful primary screening tool test for detecting diabetes and for glucose estimation in non-diabetic patients. Use of gingival crevicular blood for screening is an attractive way of identifying a reasonable option of finger-stick blood glucose measurement under the appropriate circumstances. Rapid assessment may precede diagnostic evaluation in diabetic as well as healthy patients with acute severe bleeding. In addition, gingival crevicular blood levels may be needed to monitor the diabetic output.

## 1. Introduction

Diabetes is known to be one of the major global epidemic diseases, significantly associated with mortality and morbidity worldwide, conferring a substantial burden to the health care system [1,2,3]. The global rate of diabetes prevalence has increased from 4.7% in 1980 to 8.5% in 2014, rising to 9.3% (463 million people) in 2019 [1]. Projections indicate that the total prevalence of diabetes will result in about 578 million cases by 2030 and 700 million by 2045, and will be prevalent in low- and middle-income countries [4]. Therefore, early detection of diabetes in its incipient stages is a health care priority. Appropriate screening devices and standardized methods are crucial to prevent this potentially inauspicious life condition [5,6]. Traditionally, detection of diabetes, relies primarily on the estimation of blood glucose concentration, carried out through the capillary blood glucose testing using a reflectance blood glucose meter, succeeded by diagnostic confirmatory tests (fasting plasma glucose (FPG) and/or an oral glucose tolerance test (OGTT) with a 2-h plasma glucose level of 200 mg/dL (11.1 mmol/L) applying standard criteria to find out the diagnosis of diabetes [7]. An FPG ≥ 126 mg/dL (7.0 mmol/L) requires a retest to confirm the diagnosis. Glycated hemoglobin (HbA1c) is the ‘gold standard’ diagnostic and monitoring technique for metabolic control. Although the finger-prick method is accepted as a non-invasive strategy for direct and accurate measurement of blood glucose, it has been reported that often patients are psychologically opposed to receiving the test. Previous cross-sectional studies have shown that gingival bleeding may provide an objective diagnostic method for early detection of diabetes, as part of routine developmental surveillance. The gingival bleeding is a characteristic sign of periodontal inflammation that shows a favorable profile for use as a blood glucose test [8,9,10,11,12,13,14]. Interestingly, the presence of an inflammatory lesion of the periodontium is detected through bleeding determination providing an objective diagnostic method for periodontal disease. Particularly, bleeding may be the clinically apparent source to estimate the blood glucose level. Stein et al. [15], in 1969, documented the initial observations about a potential routine source in clinical detection of diabetes, proposing the gingival blood test in estimating the blood glucose level. The authors showed that using gingival blood was a suitable and secure method in collecting blood but failing to replace the conventional glucose testing methods. Based on literature evidence [11,14,15,16,17,18], our study aimed to compare gingival crevicular blood with finger-stick blood glucose measurements using a self-monitoring glucometer, to evaluate whether gingival crevicular blood could be an alternative to allow accurate chairside glucose testing.

## 2. Materials and Methods

### 2.1. Study Design

A cross-sectional comparative study was carried out among diabetic and systemically healthy patients. The subsection of inclusion criteria was the following: (I) type 2 diabetes mellitus determined by at least one of the fasting plasma glucose ≥ 126 mg/dL; (II) random plasma glucose ≥ 200 mg/dL; (III) hemoglobin A1c (HbA1c) between 7.5 and 11.0%, (IV) use of antidiabetic medications; and (V) positive to bleeding on probing. Patients were ineligible for participation if any of the following conditions existed: a diagnosis of any malignancy, history of organ transplant, diagnosis of dementia, cardiovascular diseases, other systemic or concurrent diseases, and pregnancy were excluded from the study [4,6,7]. There was a total of one-hundred-forty patients aged 31–67 years, with a mean age (± SD) of 47.72 ± 9.47 for the diabetic group, and a mean (± SD) of 48.83± 11.97 for the non-diabetic group.

Before the glucose level estimation, a standardized clinical periodontal examination was performed employing a mouth mirror and a Michigan type ‘O’ periodontal probe at six sites on each tooth (mesiobuccal, midbuccal, distobuccal, mesiolingual, midlingual, and distolingual), by a single examiner, to detect the bleeding gingival site offering the profuse bleeding for comparison of gingival crevicular blood glucose (GCBG) sample collection. Periodontal probing was carried out under cotton roll isolation and after air drying with an air syringe. Lengths of 1–2 mm bore glass capillary tubing were used as blood receiver tubes before being transferred to the preloaded test strip into the glucometer (Accu-Chek Active, Roche Diagnostics, Indianapolis, IN, USA), which needs only approximately 5 s to report blood glucose measurements. Then, the corresponding values were recorded. Therefore, at the same visit, a simultaneous capillary finger-prick was performed and analyzed using a single calibrated and validated chairside glucometer, by an accredited registered nurse. Using the device, a finger was pricked to release a small sample of blood into the test strip, which was then analyzed by the glucometer that outputs the blood glucose reading. The glucometer is an electrochemical system for blood glucose measurement, based on an amperometric enzyme electrode principle. The system is provided with dry reagent test strips that are calibrated to report plasma glucose values. After the combination of the glucose in the blood sample with the enzyme glucose dehydrogenase, there is the conversion to gluconolactone, which generates an electrical charge. Then, the charge is measured by the electrodes incorporated in the test strip, and a digital read-out is given.

### 2.2. Governance and Ethics

Ethical approval for this study was obtained from the Ethics Committee Approval INTL_ALITSHCOOP/DentPath/2020_SLK and was conducted according to Good Clinical Practice and the Declaration of Helsinki [19]. Written informed consent was obtained from all patients before the study.

### 2.3. Statistics

For each measurement, the mean value and standard deviation (SD) were extracted. Pearson’s product–moment correlation analysis was applied to assess the associations between the studied variables. Differences in mean values between groups were analyzed using the independent Student’s two-tailed *t*-test for normally distributed data. All statistical procedures were performed at a level of significance of 5% (*P* < 0.05). All data were examined using SPSS for Windows version 11 (SPSS Inc., Chicago, IL, USA).

## 3. Results

In our study, 140 patients in total were enrolled: 70 diabetic patients, 70 healthy patients, in the age range of 31–67 years. The mean and standard deviations values for the GCBG and peripheral finger-stick blood glucose (PFBG) in the diabetic and non-diabetic groups were assessed, as shown in Table 1.

Among the diabetic patients, the mean of gingival crevicular blood glucose level was 160.42 ± 31.31 mg/dL, and the mean of capillary blood glucose level was 161.64 ± 31.56 mg/dL, and *t*-value was found to be 0.185. Among the non-diabetic group, the mean of gingival crevicular blood glucose level was 93.93 ± 20.93 mg/dL, and the mean of capillary blood glucose level was 90.88 ± 19.38 mg/dL. The *t*-value was found to be 1.127. Pearson’s correlation analysis revealed a strong positive correlation between capillary blood glucose level and gingival crevicular blood glucose (*r* = 0.9834, *P* < 0.0001) for the diabetic group, and *r* = 0.8153, *P* < 0.0001 for the non-diabetic group, respectively (Table 2).

Figure 1 illustrates the comparison between gingival and capillary blood glucose among the diabetic group.

Figure 2 shows the comparison between gingival and capillary blood glucose among the non-diabetic group. On comparison, no significant indication of the similarity between both test levels was found. No statistically significant difference was found between the two values for both groups.

## 4. Discussion

Epidemiological data derived from the National Diabetes Statistics Report (2020) indicate that 7.3 million people (21.4%) have undiagnosed diabetes in the United States. Currently, there are 425 million people affected by diabetes, and this number is expected to rise by 48% worldwide. The multifaceted nature of diabetes poses a challenge to early identification of the disease. The screening process is the critical first step of diabetes risk surveillance and management, playing an integral and increasing role in supporting public health responses to this urgent health event. By definition, the rationale for early detection of diabetes is that this process might permit the presumptive identification of at-risk patients, prior to the development of hard endpoints. The primary purpose of screen action aiming is to identify an existing disease in an asymptomatic population and to improve outcomes. In view of the above, the capillary blood glucose testing using a reflectance blood glucose meter is the commonly used and validated screening tool for the determination of glucose levels. Despite its proven effectiveness and utility for the assessment of glycemic status, this method remains uncomfortable, potentially limiting the compliance [8,9]. Gingival crevicular blood has been suggested as a reliable glucose measurement and acceptable screening source. In this view, our study aimed to evaluate whether gingival crevicular blood may be an effective presumptive test that allows identification, or confirmation, of the presence of diabetes in an apparently healthy, asymptomatic population, assuming that gingival bleeding might have a predictive value for diabetes. Interestingly, for two decades, researchers have sought evidence of a positive correlation between periodontal infection and diabetes. Specifically, the adverse impact of the structural and functional alterations in diabetes affects the periodontium, which provokes a large spectrum of clinical manifestations of periodontal illnesses with the gingival bleeding as a peculiar sign and symptom. Diabetic patients often have a co-occurring periodontal infection, closely related to poor glycemic control. For this reason, undiagnosed diabetes may be more likely encountered during a periodontal examination to refer for a diagnostic test later. An argument can be put forward that the potential screening ability of GCB is becoming increasingly important with the recognition of the relationship between diabetes and periodontal disease. The debate surrounding the possible utilization of GCB for screen diabetes has been heightened by research on the impact of diabetes on periodontal disease and vice versa. A number of recent studies have reported similar findings, with measurement differences reduced or even eliminated. Previous studies have compared the GCBG value with the PFBG value to determine whether gum’s sulcular blood could offer an option to determine the blood glucose level, balancing the composite issue involved in the choice made by clinicians. Findings revealed strong positive correlations among capillary blood glucose levels with gingival blood glucose levels. Our data extend previous findings using the same study design that have documented successful blood glucose evaluation in undiagnosed diabetic patients and diabetic population. Interest in the detection of diabetes using GCB has been heightened by the results of several trials for the treatment of periodontal patients [7,12,13,14,15,16,17,18,19,20,21,22,23,24,25,26,27,28,29,30,31,32,33,34]. Of course, the optimal condition for GCB measurement occurs when the bleeding is abundant. Interestingly, the study conducted by Strauss et al. [5] specified that gingival crevicular blood samples were appropriate to screen for diabetes in individuals suffering from copious gingival bleeding to acquire a sample. Early transition to an optimal periodontal condition interferes with the blood collection procedure, impeding the monitoring phase [28,31]. Furthermore, any adulteration of the collected blood sample by saliva and oral debris may provide an alteration of the results. For this reason, to collect samples, we isolated the bleeding gingival site with cotton rolls. Some studies have been conducted comparing the venous blood glucose concentration with gingival crevicular blood sugar in order to evaluate the performance of GCB as a non-invasive screening tool. Results reported by Sarlati et al. [23], registered a strong correlation between diabetic venous blood glucose concentration (VFBS) and gingival crevicular fasting blood sugar (GFBS) (*r* = 0.962, *P* = 0.000). Parihar et al. extended the previous findings by demonstrating the potential of gingival blood for developing an alternative measure of diabetes screening as a supplement to the current method, performing the comparison of GCB source for glucose blood estimation with two standard screening methods: finger-stick blood (FSB) and venous blood (VB). The authors reported a strong positive correlation between glucose levels of GCB with finger-stick blood (FSB) and venous blood (VB) with values of 0.986 and 0.972 in the diabetic group and 0.820 and 0.721 in the non-diabetic group. This study was consistent with the findings of Patil and Kamalakkannan [24] that showed a correlation among these measurements (*r* = 0.736, *P* = 0.001). Recent evidence of the validity of GCB in detecting diabetes derives from the study conducted by Sibyl et al. [25], which performed an analytical study, collecting blood samples by the finger-stick method and periodontal probing in order to correlate the measurements of the two methods. These results go beyond previous reports, showing a high correlation (*r* = 0.97) of these two tests for patients presenting adequate bleeding on probing. The authors concluded that gingival crevicular blood testing might be contemplated to increase the screening yield of diabetes in routine dental clinical practice. In contrast, few studies failed to demonstrate the validity of gingival crevicular blood as a screening tool in diabetes mellitus, reporting consistent differences between the finger-prick blood (FPB) and gingival crevicular blood screening tool measurements. Kandwal and Batra [26] registered a very low positive correlation with *r* = + 0.045 for GCB and FPB for the diabetic group and *r* = + 0.0324 for the non-diabetic group.

In accordance with previous findings, our results highlight that the estimation of sulcular blood glucose levels shows a correlation with capillary blood glucose levels, thereby advocating that testing sulcular blood may be an advantageous tool in detecting potential patients with diabetes. Benefits of GCB glucose measurement include the ability to perform the test at the chairside and do not require a relatively long time for results. Furthermore, the procedure can be performed by a dentist. However, there is also the limited applicability as a routine test because the gingival blood source availability is closely tied with the degree of tissue’s inflammatory status. Indeed, elevated bleeding is often associated with a high level of tissue inflammation and vice versa. Consequently, after periodontal treatment, the improvement of the inflammatory status determines a reduction of bleeding levels, leading to a minor availability of the gingival blood source for measurement of glucose level. The glucometric system requires a blood volume of 4 μL.

## 5. Conclusions

The blood glucose measurement is generally performed using a self-monitoring glucometer. Our study aimed to investigate whether the gingival blood glucose might be a source for glucometric analysis during the dental office visit. The use of gingival crevicular blood (GCB) is appealing owing to its rapidity and extensive availability, the potential for rapid assessment, and having an additional benefit in patients with diabetes by reducing the stress. Furthermore, GCB should be used in dental routine after periodontal evaluation. Despite the intuitive advantages, gingival crevicular blood has not proven superior to the capillary finger-stick test. The major consideration is that the presence of bleeding has the potential to be a realistic alternative to screen diabetes in selected patients, but longitudinal studies are needed to confirm the effectiveness of disease detection. In considering the developments in screening methods, the exploration and consideration of GCB as an alternative screening tool is still needed. Glucometer performed in the dental office may be an important device in diagnosing diabetes so it can be addressed during a periodontal examination, for its simplicity, cheapness, and minimum disturbance for individuals [11,12,13]. In addition, gingival crevicular blood levels may be needed to maintain an adequate diabetic output [15,18,19,20,21]. Therefore, there is also the limited applicability as a routine technique for the screening due to its requirement of a minimum blood volume.

## Figures and Tables

**Figure 1 ijerph-17-07356-f001:**
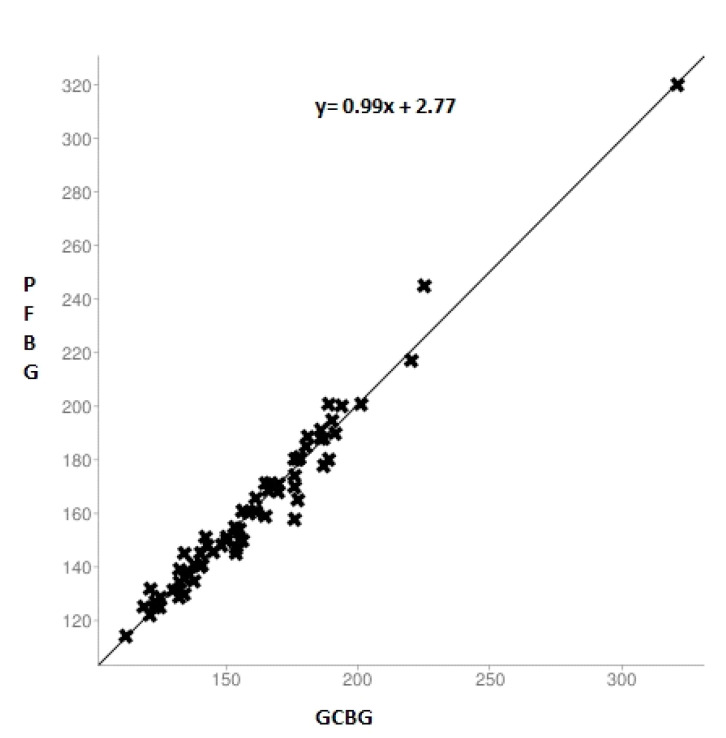
Scatter plot of the linear relationship between the glucose measurements of the gingival blood and capillary finger-stick method in the diabetic group. GCBG: comparison of gingival crevicular blood glucose.

**Figure 2 ijerph-17-07356-f002:**
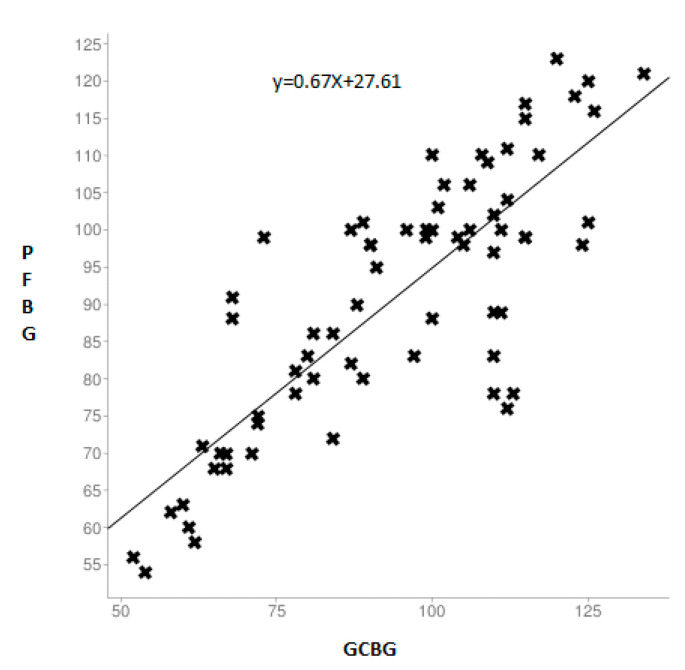
Scatter plot of the linear relationship between the glucose measurements of the gingival blood and capillary finger-stick method in the non-diabetic group.

**Table 1 ijerph-17-07356-t001:** Mean and standard deviation measured in both groups

Groups	No	Statistical Data	GCBG (mg/dL)	PFBG (mg/dL)	*P-*Value *	*T-*Value *
Diabetic	70	Mean ± SD	160.42 ± 31.31	161.64 ± 31.56	0.9834	0.185
Non-diabetic	70	Mean ± SD	93.93 ± 20.93	90.88 ± 19.38	0.8153	1.127

* *t*-value and *P*-value for GCBG compared to PFBG in both groups; GCBG—comparison of gingival crevicular blood glucose; PFBG—peripheral finger-stick blood glucose.

**Table 2 ijerph-17-07356-t002:** * Karl Pearson’s product–moment correlation (*r*) for gingival crevicular blood glucose values and peripheral finger-stick blood glucose values in the diabetic and the non–diabetic group, respectively.

Group	* Correlation (*r*)
Diabetic	0.9834
Non-diabetic	0.8153

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
