# Peer review of "Gingival Crevicular Blood as a Potential Screening Tool: A Cross Sectional Comparative Study"

_ijerph, 2020, doi:10.3390/ijerph17207356_

Round 1

Reviewer 1 Report

In this cross-sectional comparative study, the authors studied the correlation between capillary blood glucose and gingival crevicular blood glucose in diabetic and non-diabetic patients.  They suggested that gingival crevicular blood for diabetic screening could be an option of finger stick blood glucose measurement.  Overall, this study is interesting and may be of clinical use.

There are several concerns regarding to this manuscript.

-What are the ages (Mean ± SD) for the diabetic or non-diabetic groups? Is there any correlation between age and the health condition of the gum? If so, will this affect the gingival crevicular blood collection among patients at different ages?

-What is the ratio between male and female patients?

-What is the limitation for the gingival crevicular blood collection?  Is there any difference between diabetic and non-diabetic groups, such as the feasibility or the source availability?

-What is the reproductively of this gingival crevicular blood glucose measurements?

-What is the cutoff for diabetes if gingival crevicular blood is used for diagnosis?

Author Response

#Reviewer 1

  • What are the ages (Mean ± SD) for the diabetic or non-diabetic groups? Is there any correlation between age and the health condition of the gum? If so, will this affect the gingival crevicular blood collection among patients at different ages?

-The mean Mean ± SD has been added at line 92, as follows: 47.72 ± 9.47 for diabetic group, and 48.83± 11.97

- A positive correlation between age and the health condition of the gum is reported in literature, but no linear correlation can be concluded.

  • What is the ratio between male and female patients?

-Studies investigating the differences in periodontal parameters of diabetic patients after gender subdivision, report that male with type 2 diabetes have significantly more periodontal worsen condition compared to diabetic female (p < 0.05), higher HbA1c values (p < 0.001), and significant higher periodontal indices values (p < 0.04). Women described a better oral hygiene care and differences in gum’s inflammation were significantly lower than those of men.

  • What is the limitation for the gingival crevicular blood collection?  Is there any difference between diabetic and non-diabetic groups, such as the feasibility or the source availability?

-The limitation for the gingival crevicular blood collection has been indicated at line 223-228, corrected as follows:

“…there is also the limited applicability as a routine test because the gingival blood source availability is closely tied with the degree of tissue’s inflammatory status. Indeed, elevated bleeding is often associated with high level of tissue’s inflammation and vice versa. Consequently, after periodontal treatment, the improvement of the inflammatory status determines a reduction of bleeding levels, leading to a minor availability of the gingival blood source for measurement of glucose level. The glucometric system requires a blood volume of 4 μL”

  • What is the reproductively of this gingival crevicular blood glucose measurements?

- As aforementioned, the reproductively of this gingival crevicular blood glucose measurements it’s a limitation because the availability of minimum blood amount depends on the degree of tissue’s inflammatory status. Indeed, elevated bleeding is often associated with high level of tissue’s inflammation and vice versa. Consequently, after periodontal treatment, the improvement of the inflammatory status

  • What is the cutoff for diabetes if gingival crevicular blood is used for diagnosis?

-It is not possible to determine the cutoff for diagnosis.  As described at lines 61-68, the gingival crevicular blood, as well as the capillary finger blood, cannot be used for diagnosis because it is only an extemporaneously determination of capillary blood glucose level, useful for monitoring Diabetes. The WHO recommends using both the fasting plasma glucose and the oral glucose tolerance test with a 2-hour plasma glucose level of 200 mg/dL (11.1 mmol/L) to establish the diagnosis of type 1 and type 2 diabetes.

Reviewer 2 Report

I am grateful for the possibility to revise this research study.

Gingival crevicular blood as a potential screening tool is a trend topic in the current research literature and may be a main focus of interest for readers.

On the other hand blood screening is a useful medical procedure and management of this technique could reduce the complication rates.

 The title is appropriate

The abstract sections reflect adequate the main objective of study

Introduction may be improved adding new information in order to provide an adequate state-of-the-art including some references.

Methods are well-designed with relevant and complete information. Correct search strategies, good description of the process however there not detailed statistical analyses were included. In order to analyse the level evidence of the results obtained in the research

Please, add references to cite the exclusion criteria

The authors have investigated using gingival crevicular blood. There are several methodological concerns that limit the reader's understanding of why this experiment was conducted. Below I have provided comments for the authors.

What is a point to measure compare gingival crevicular blood ?

Furthermore, please provide a hypothesis.

Tables, figures and redaction of the results are presented in confused way. Authors should justify better their results I suggest authors may improve this aspect for example, should include differences by sex and if it is possible significate differences between different diagnostics

Discussion section include future research studies secondary to the current findings of this study. Clinical considerations, limitations and overall discussion are well-presented

Author Response

#Reviewer 2

  • Introduction may be improved adding new information in order to provide an adequate state-of-the-art including some references.
  • Few studies have been conducted recently about our objective. The limited references are due to the impossibility to report more current investigations
  • Please, add references to cite the exclusion criteria

-References have been added as follows:

  1. Ardakani, M.R.; Moeintaghavi, A.; Haerian, A.; Ardakani, M.A.; Hashemzadeh, M. Correlation between levels of sulcular and capillary blood glucose. J Contemp Dent Pract. 2009, 10, 10–7.
  2. Kempe, K.C.; Budd, D.; Stern, M.; Ellison, J.M.; Saari, L.A.; Adiletto, C.A.; Olin, B.; Price, D.A.; Horwitz, D.L. Palm glucose readings compared with fingertip readings under steady and dynamic glycemic conditions, using the OneTouch Ultra Blood Glucose Monitoring System. Diabetes Technol Ther. 2005, 7, 916–26.
  3. Müller, H.P.; Behbehani, E. Screening of elevated glucose levels in gingival crevice blood using a novel, sensitive self-monitoring device. Med Princ Pract. 2004, 13, 361–525.
  • The authors have investigated using gingival crevicular blood. There are several methodological concerns that limit the reader's understanding of why this experiment was conducted. Below I have provided comments for the authors.

    What is a point to measure compare gingival crevicular blood?

-We aimed to investigate if the both measurement tests could determine the same value, without a specific point of measure. The standard measure was mg/dL, as specified in the test.

  • Furthermore, please provide a hypothesis.
  • - the use of the gingival crevicular blood provides the same values of blood glucose level of capillary blood measurement”, the null hypothesis is “the gingival crevicular blood doesn’t provide the same values of blood glucose level of capillary blood measurement.”

  • Tables, figures and redaction of the results are presented in confused way. Authors should justify better their results I suggest authors may improve this aspect for example, should include differences by sex and if it is possible significate differences between different diagnostics

-As aforementioned, our focus was to compare the measurement of blood glucose level using gingival crevicular blood with blood glucose level using the capillary blood. We aimed to investigate the plausibility of using the gingival crevicular blood focusing only on the correspondence of levels measured through the two different methods. We aimed to determine if the level of N mg/dL (for example 125 mg/dL) revealed with the finger capillary measurement could be correspondent to the measurement assessed with gingival crevicular blood (as for the example, 125 mg/dL).

Round 2

Reviewer 1 Report

Most of the reviewer's questions were properly addressed.